# Magnitude and factors associated with iron supplementation among pregnant women in anemia hot spot regions of Ethiopia: Multilevel analysis based on Bayesian approach

**Yilkal Negesse** [1]*, **Habtamu Temesgen**[1], **Wubetu Woyraw**[1], **Melsew Setegn Alie**[2], **Ayenew Negesse**[1]

1 College of Health Sciences, Debre Markos University, Debre Markos, Ethiopia, 2 College of Medicine and Health Sciences, Mizan Tepi University, Mizan Aman, Ethiopia

* negeseyilkal@gmail.com

**Data Availability Statement:** The data for this study were obtained from the Demographic and Health Surveys (DHS), which can be accessed at

## Abstract

### Background

According to World Health Organization, pregnant women should take an oral iron and folic acid every day for at least 3 months to prevent preterm birth, low birth weight, maternal anemia, and puerperal sepsis. In addition to keeping maternal health, it also plays a key role to support the fetus's healthy growth and development. Therefore, it is very important to know the magnitude of iron supplementation and its determinants in anemia hot spot regions of Ethiopia using an appropriate statistical analysis method.

### Objective

The aim of this study is to determine the magnitude of iron supplementation and its associated factors in anemia hot spot regions of Ethiopia among pregnant women.

### Methods

The study was done using the 2019 Ethiopian Mini Demographic and Health surveys data. Before any statistical analysis was done, the data were weighted using sampling weight for probability sampling and non-response. Then, a total weighted sample of 2116 reproductive age group women in anemia hot spot regions of Ethiopia were used for this study. A multilevel binary logistic regression model based on the Bayesian approach was fitted using the **Brms** R package to identify the determinants of iron supplementation in anemia hotspot regions of Ethiopia. Finally, the 95% credible interval (CrI) of the adjusted odds ratio (AOR) was used to assess statistical significance. If the interval includes 1, the result is considered non-significant.

the following link http://dhsprogram.com/data/available-datasets.cfm.

**Funding:** The author(s) received no specific funding for this work.

**Competing interests:** The authors have declared that no competing interests exist.

**Abbreviations:** AOR, Adjusted Odds Ratio; C, Difference due to coefficient; CrI, Credible interval; DHS, Demographic Health Survey; E, Difference due to characteristics; EA, Enumeration Area; EDHS, Ethiopian Demographic and Health Survey; HMC, Hamiltonians Monte Carlo; L-CrI, Lower Credible Interval; LOO, Leave-One-Out Cross-Validation; SE, Standard Error; U-CrI, Upper Credible Interval.

## Results

This study showed that in anemia hotspot regions of Ethiopia, the overall magnitude of iron supplementation among pregnant women is 55.5% (95% CrI: 53.4%- 57.6%). Being rural resident (AOR = 0.57; 95% CrI 0.34–0.93), having higher education level (AOR = 3.2; 95% CrI 1.80–5.7), having secondary education level (AOR = 3.28; 95% CrI 2.13–5.1), being wealthy (AOR = 1.80; 95% CrI 1.27–2.54), being household headed by female (AOR = 0.55; 95% CrI 0.43–0.71) and, have no children (AOR = 0.4;95%CrI 0.17–0.98) were significantly associated with iron supplementation among pregnant women in anemia hotspot regions of Ethiopia.

## Conclusion

The overall magnitude of iron supplementation among pregnant women in anemia hotspot regions of Ethiopia is notably low when compared to the World Health Organization's recommended target. Significant factors associated with higher iron supplementation included having secondary or higher education, rich in wealth, and being from a male-headed household. Conversely, being a rural resident, female-headed household, and having no children were associated with lower iron supplementation.

## Background

Pregnant women are advised to take an oral iron and folic acid every day by the World Health Organization (WHO) to prevent maternal and perinatal mortality [1, 2]. The recommendation aims to prevent disorders including preterm birth, low birth weight, maternal anemia, and puerperal sepsis [1–4]. It also aims to meet the increasing need for folic acid, iron, and other vital nutrients during pregnancy in order to support the fetus's healthy growth and development [1–4].

In females, a hemoglobin level of less than 12 g/dL is deemed abnormal and indicates anemia. However, anemia during pregnancy is defined as a hemoglobin level that is below 11 g/dL at any time [5]. Iron deficiency anemia is highly prevalent, particularly in developing countries, and it is becoming an international health problem [6, 7]. Over 2 billion people worldwide suffer from iron deficiency [8]. Around 52% of pregnant women in developing countries experienced the negative effects of iron deficiency [9]. Specifically, Sub-Saharan Africa (SSA) is the most affected region by this health condition [10].

Due to a decreased oxygen supply to the tissues, iron deficiency anemia can cause a variety of symptoms in the women, including fatigue, paleness, apathy, fainting, and difficulty breathing [9, 11]. Headaches, palpitations in the heart, hair loss, and ringing in the ears are possible additional symptoms [9]. In extreme situations, heart failure could happen. It also increases the risk of premature labor, impaired fetal growth, low birth weight, birth asphyxia, and neonatal anemia.

So, the World Health Organization recommends frequent maternal iron supplementation as a means of resolving the global problem of iron deficiency anemia and preventing its negative consequences [12, 13]. However, it is concerning that numerous women complete their pregnancies without meeting the minimum recommended iron intake [14]. The percentage of pregnant women in Sub-Saharan Africa who took iron supplements during their pregnancy was reported to be 28.7%. However, the figure differed greatly throughout countries, with

Burundi having the lowest rate at 1.4% and Senegal having the highest percentage at 73.0% [15].

The study conducted in Ethiopia identified specific regions that have a higher incidence of anemia. Among the nine regional states and two city administrations, the study found that five regions, namely Tigray, Afar, Oromia, Somalia and Gambela regional states were hotspot regions for anemia [16, 17]

Even though iron deficiency anemia is associated with maternal and children morbidity and mortality, scholars missed to determine the magnitude of iron supplementation and its associated factors in anemia hotspot regions of Ethiopia using an appropriate statistical method of analysis. So, it is very important to determine the magnitude of iron supplementation and its determinants in anemia hotspot regions of Ethiopia using an appropriate statistical method of analysis. Therefore, this study was conducted to determine the magnitude of iron supplementation and its determinants in anemia hotspot regions of Ethiopia based on the Bayesian statistical method of analysis.

## Methods and materials

### Data source and population

To conduct this study, we used data from the 2019 Ethiopian Mini Demographic and Health Survey (EDHS). This dataset contains information regarding various characteristics of children, women, and men. From March to June 2019, the data was collected from the nation's two city administrations, and all nine regional states [18]. A two-stage stratified sampling technique was used to collect the data. Each region was divided into rural and urban areas, resulting in a total of 21 sampling strata. In the initial stage, a random selection of 305 Enumeration Areas (EAs) was made, and subsequently, an average of 30 households per EA were chosen [18–21].

After permission was secured through an online request by explaining the objective of our study, the data was accessed from the Measure DHS website (http://www.dhs). For this study, we utilized the Individual Recode (IR) file, which contains data on women of reproductive age (15–49 years). Out of the 8,885 women in the IR file, 3,908 were eligible to answer questions related to iron supplementation, while 4,977 were not. However, research focused specifically on the anemia hotspot regions of Ethiopia. After excluding mothers who didn't know whether they took iron supplements or not, the final sample size becomes 2,116 women in the anemia hotspot regions.

### Variables

The outcome variable of this study was iron supplementation during pregnancy. The response is categorized as "Yes" or "No" and coded as "Yes = 1" and "No = 0". The EDHS asked respondents to answer the question "did you received iron tablets or syrup for 90 days during your recent pregnancy?". So, the response is dichotomous with possible values $Y_i = 1$, if the $i^{th}$ women received iron tablets or syrup for 90 days and $Y_i = 0$, if the $i^{th}$ women don't receive it for 90 days during her recent pregnancy.

The independent variables were classified as a community and individual-level variables. The place of residence and region of the study participants were considered as community-level variables. Whereas age of the women, educational status of the women, marital status of the women, sex of household head, wealth status, number of births in the last five years and, total number of living children, were considered as individual-level factors.

## Data processing and analysis

After downloading the data from the Measure DHS website, the datasets were extracted from the Individual Record (IR) file and subsequently cleaned and coded using Stata version 14.2 software. Then, in order to ensure representativeness of the survey and obtain reliable statistical estimates, the data were weighted using sampling weights. During in the analysis, two level structure of the data was considered. All pregnant women within the household were considered as first level and enumeration areas where pregnant women are nested are considered as second level. After the data were cleaned and coded, descriptive statistical analysis was done using STATA version 14.2 and multilevel binary logistic regression analysis was done based on the Bayesian approach to see the relationship between the dependent and independent variables. To fit multilevel binary logistic regression model and to estimate the parameters of the variables and the level of random variations between clusters **Brms R-package** was used. In Bayesian approach, the estimates of the parameters are obtained from the posterior distribution, which is the combination of the likelihood of the data and prior information [22]. To estimate the variance and regression coefficients, we considered a vague prior with beta (1, 1) and gamma (0.001, 0.001) distribution respectively. Then, we used, 2 chains, 8,000 iterations, 0 as the starting values of the iterations, 1000 = iterations that was discarded, adapt delta = 0.95 to control the divergent transition, = 1000, initials and cores = 2.

To simulate direct draws from the complex posterior distribution, we employed the No-U-Turn Sampler (NUTS) method. This method avoids the random walk behavior and sensitivity to correlated parameters that are problematic in many Markov Chain Monte Carlo (MCMC) methods. NUTS takes a series of steps informed by first-order gradient information, allowing for more efficient and accurate sampling [23].

Four models were fitted by using multilevel binary logistic regression. Then, based on their Leave-One-Out Cross-Validation (LOO) and Widely Applicable Information Criteria (WAIC) value each fitted model was compared and a model which have a small WAIC and LOO value was selected as a best-fitted model. The selected model incorporates both individual-level and community-level variables, allowing for simultaneous adjustment of potential confounders while examining the association between iron supplementation and independent variables. After that, all interpretations and inferences were made based on the selected model (a model which have small LOO and WAIC value). To consider a variation in magnitude iron supplementation across clusters (EA) in anemia hot spot regions of Ethiopia, we used the ICC value greater than 10%. Finally, the 95% posterior credible interval which doesn't include 1 was considered to declare statistically significant association between independent and dependent variables.

In Bayesian analysis, before reporting the results, it's better to check the reliability of the findings by assessing the convergence of the Algorithms. Because, unless the chain of the posterior distribution reached its stationary point, the results obtained from a given HMC analysis are not deemed reliable [24]. Therefore, to monitor the convergence of the algorithm, we used Tail_ESS and Bulk_ESS were greater than 1000, Rhat = 1, chains of the time serious plots were mixed well, and smooth density plot.

## Ethics approval

Ethical approval was obtained from the Measure Demographic and Health Survey (DHS) by completing the data access request form (http://www.dhsprogram.com). The data utilized in this study consist of freely available aggregated secondary data, which do not contain any personal identifiers. The requested data were used strictly in an anonymous manner and

**Table 1. Frequency and percentage distribution of respondents in anemia hot spot regions of Ethiopia.**

| Variables | Categories | Frequency | Percent |
|---|---|---|---|
| Residence | Urban | 485 | 22.9 |
| | Rural | 1631 | 77.1 |
| Regions | Tigray | 285 | 13.5 |
| | Afar | 51 | 2.4 |
| | Oromia | 1511 | 71.4 |
| | Somalia | 217 | 10.3 |
| | Gambela | 19 | 0.9 |
| | Harar | 11 | 0.5 |
| | Dire Diwa | 21 | 1 |
| Age group | 15–25 | 868 | 47.4 |
| | 26–35 | 934 | 44.1 |
| | 36–49 | 314 | 14.8 |
| Educational level of the mother | No formal education | 1080 | 51.1 |
| | Primary | 779 | 36.8 |
| | Secondary | 187 | 8.9 |
| | Higher | 69 | 3.2 |
| Marital status | Married | 1958 | 92.5 |
| | Single | 26 | 1.2 |
| | Separated | 39 | 1.9 |
| | Widowed | 30 | 1.4 |
| | Divorced | 63 | 3 |
| Wealth index | Poor | 1002 | 47.4 |
| | Medium | 378 | 17.9 |
| | Rich | 736 | 34.8 |
| Sex of household head | Male | 1550 | 73.2 |
| | Female | 566 | 26.8 |
| Births in last five years | < = 2 | 1986 | 94.2 |
| | > = 3 | 121 | 5.8 |
| Number of living children | No | 27 | 1.3 |
| | 1–3 | 1103 | 52.1 |
| | > = 4 | 986 | 46.6 |

exclusively for research purposes. Comprehensive information regarding ethical considerations is detailed in the EDHS report [25].

## Result

### Characteristics of the study participants

Table 1 reveals that a lower percentage of participants (22.9%) live in urban areas, whereas a considerable majority of participants (77.1%) live in rural areas. When we see the regional distribution, it becomes apparent that the Oromia regional state had the highest number of respondents (71.4%), followed by Tigray (13.5%) and Somalia regional state (10.3%).

Furthermore, a significant observation can be made regarding the age groups of the respondents. Nearly half of the participants belong to the age group of fifteen to twenty-five years, with a close second being the age group of twenty-six to thirty-five years. This information highlights the prominence of young individuals within the surveyed population.

**Table 2. Frequency and percentage distribution iron supplementation among pregnant women based on different characteristics in anemia hotspot regions of Ethiopia.**

| Variables | Categories | Frequency | Percent |
|---|---|---|---|
| Residence | Urban | 308 | 63.5 |
| | Rural | 867 | 53.2 |
| Regions | Tigray | 242 | 86 |
| | Afar | 27 | 52 |
| | Oromia | 833 | 55.1 |
| | Somalia | 41 | 19 |
| | Gambela | 12 | 58 |
| | Harar | 8 | 66.7 |
| | Dire Diwa | 15 | 68.2 |
| Age group | 15–25 | 521 | 60 |
| | 26–35 | 520 | 55.7 |
| | 36–49 | 135 | 43 |
| Educational level of the mother | No formal education | 436 | 40.3 |
| | Primary | 520 | 66.7 |
| | Secondary | 159 | 85 |
| | Higher | 61 | 88.4 |
| Marital status | Married | 1087 | 55.5 |
| | Single | 14 | 53.8 |
| | Separated | 24 | 60 |
| | Widowed | 10 | 30 |
| | Divorced | 42 | 66.7 |
| Wealth index | Poor | 425 | 42.4 |
| | Medium | 240 | 63.5 |
| | Rich | 511 | 69.4 |
| Sex of household head | Male | 886 | 57.2 |
| | Female | 289 | 51.1 |
| Births in last five years | < = 2 | 1130 | 57 |
| | > = 3 | 43 | 34.7 |
| Number of living children | No | 10 | 37 |
| | 1–3 | 721 | 65.4 |
| | > = 4 | 444 | 45 |

## The magnitude of iron supplementation among pregnant women in anemia hotspot regions of Ethiopia

In Ethiopia's anemia hotspot regions, 55.5% of pregnant women (1,175 out of 2,116) receive iron supplementation, with a 95% CrI: 53.4%- 57.6%. According to Table 2, 63.5% of pregnant women living in urban areas and 53.2% of pregnant women in rural areas, respectively, received iron supplementation. In terms of regional distribution, it is clear that, at 86%, the Tigray region recorded the highest proportion of pregnant women receiving iron supplementation. About 52% of pregnant women in the Oromia region and 55.1% of pregnant women in the Afar region received this intervention. However, with only 19% of pregnant women receiving this supplementation, the Somalia region had the lowest proportion of iron supplementation. These figures highlight the regional disparities in access and implementation of iron supplementation programs for pregnant women.

In the age group of 15–25, 60% of pregnant women received iron supplementation, indicating a relatively high proportion. Moving to the age group of 26–35, the percentage slightly decreased to 55.7%, suggesting a slightly lower uptake of iron supplementation among this age group. Among pregnant women aged 36–49, the proportion of iron supplementation dropped further to 43%. The table also shows that 55.5% of married pregnant women received iron supplementation. Comparably, 53.8% of pregnant women who were single received iron supplements. However, just 30% of pregnant widows received iron supplementation, which is a lower percentage.

## Multilevel analysis based on Bayesian approach

The final model, which incorporates both individual and community-level variables, showed the best fit among the four multilevel models fitted using the Bayesian technique, as indicated by the least WAIC (Widely Applicable Information Criterion) value.

In Bayesian analysis, the convergence of the Markov Chain Monte Carlo (MCMC) algorithm is assessed using the Gelman-Rubin statistic, denoted as the "Rhat" column. Convergence is considered high when the values are close to 1. The "Tail_ESS" column represents the tails of the posterior distribution, while the "Bulk_ESS" column represents the bulk of the distribution. Higher values in these columns indicate more reliable parameter estimates.

The MCMC algorithms for the model achieved good convergence. This is evident from the fact that the Bulk_ESS and Tail_ESS values exceed 1000, and the Rhat value equals one for all parameter estimates (Table 3).

The findings shown in Table 3 indicate that there is a significant variation (variance = 1.9) in the magnitude of iron supplementation among pregnant women in anemia hotspot regions of Ethiopia across different enumeration areas. Furthermore, as demonstrated by the Intra cluster Correlation Coefficient (ICC) value of 28%, the study found that the enumeration areas account for roughly 28% of the variation in iron supplementation among pregnant women.

Among the variables included in the final model; residence, educational level, wealth status, sex of household head, and number of living children are factors that are significantly associated with iron supplementation in anemia hot spot regions of Ethiopia. The odds of receiving iron supplementation among rural resident pregnant women is 43% (AOR = 0.57; 95% CrI 0.34–0.93) less likely as compared to women who live in urban areas. The odds of receiving iron supplementation among women who attended higher education is 3.21times (AOR = 3.2; 95% CrI 1.80–5.7) higher as compared to women who didn't have formal education. Similarly, the odds of receiving iron supplementation among women who attended secondary education is 3.28 times (AOR = 3.28; 95% CrI 2.13–5.1) higher as compared to women who didn't have formal education. The odds of receiving iron supplementation among women who have no children is 60% (AOR = 0.4;95%CrI 0.17–0.98) is less likely as compared to pregnant women who have more than four children.

## Discussion

In our study, we found that only 55.5% of pregnant women in anemia hotspot regions of Ethiopia received iron supplementation. According to the World Health Organization (WHO), all pregnant women should take daily oral iron and folic acid supplements for at least three months to prevent maternal and perinatal mortality [1, 2]. This finding highlights a significant gap, as nearly half of the target population did not receive the recommended supplementation, leaving them at risk of adverse health outcomes [1, 2].

The multilevel level binary logistic regression model fitted using the Bayesian statistical approach revealed that, among the variables included in the selected model; residence,

**Table 3. Factors associated with iron supplementation among pregnant women in anemia hot spot regions of Ethiopia.**

| Fixed effect | Characteristics | Estimates | SE | AOR | 95%CrI | | Rhat | Tail_ESS | Bulk_ESS |
|---|---|---|---|---|---|---|---|---|---|
| | | | | | L-CrI | U-CrI | | | |
| $\beta 0$-intercept | | -1.84 | 0.11 | 0.16 | 0.04 | 0.63 | 1 | 8756 | 8847 |
| Residence | Urban (ref) | | | | | | | | |
| | Rural * | -0.57 | 0.14 | 0.57 | 0.34 | 0.93 | 1 | 9876 | 8456 |
| Age group | 15-25(ref) | | | | | | | | |
| | 26–35 | 0.22 | 0.12 | 1.24 | 0.95 | 1.63 | 1 | 6123 | 7865 |
| | 36–49 | -0.12 | 0.18 | 0.90 | 0.61 | 1.32 | 1 | 8910 | 9345 |
| Educational level of Mother | No formal education (ref) | | | | | | | | |
| | Primary* | 0.68 | 0.25 | 1.97 | 1.53 | 2.53 | 1 | 7892 | 8456 |
| | Secondary* | 1.19 | 0.73 | 3.28 | 2.13 | 5.1 | 1 | 7834 | 9123 |
| | Higher* | 1.17 | 0.94 | 3.21 | 1.8 | 5.7 | 1 | 8132 | 8654 |
| Marital status | Married (ref) | | | | | | | | |
| | Single | 0.35 | 0.56 | 1.42 | 0.66 | 3.10 | 1 | 10982 | 11002 |
| | Separated | -0.07 | 0.35 | 0.93 | 0.45 | 1.93 | 1 | 10245 | 10654 |
| | Widowed* | -0.99 | 0.15 | 0.37 | 0.17 | 0.80 | 1 | 10345 | 11054 |
| | Divorced | -0.05 | 0.26 | 0.95 | 0.56 | 1.61 | 1 | 11045 | 11987 |
| Wealth index | Poor (ref) | | | | | | | | |
| | Medium* | 0.37 | 0.25 | 1.45 | 1.03 | 2.05 | 1 | 7898 | 8945 |
| | Rich* | 0.59 | 0.32 | 1.80 | 1.27 | 2.54 | 1 | 7678 | 8345 |
| Sex of household head | Male(ref) | | | | | | | | |
| | Female * | -0.60 | 0.07 | 0.55 | 0.43 | 0.71 | 1 | 8971 | 9345 |
| Births in last five years | < = 2(ref) | | | | | | | | |
| | > = 3 | -0.20 | 0.17 | 0.82 | -0.98 | 0.33 | 1 | 6889 | 7354 |
| Number of living children | No * | -0.10 | 0.18 | 0.40 | 0.17 | 0.98 | 1 | 6798 | 7279 |
| | 1–3 | 0.01 | 0.16 | 1.01 | 0.74 | 1.37 | | 7032 | 8054 |
| | > = 4(ref) | | | | | | | | |
| Random effect | | | | | | | | | |
| Variance | | 1.90 | 0.35 | | 1.33 | 1.2.72 | 1 | 8678 | 9846 |
| ICC | | 28.16 | | | 22.00 | 35.20 | | | |
| LOO | | 9214 | | | | | | | |
| WAIC | | 8126 | | | | | | | |

(Ref = reference category), ($\beta 0$ = Y intercept), (* = significant at 5% level of significance), ($\sigma_{\mu_0}{}^2$* = variance).

educational level, wealth status, sex of household head, and number of living children are factors significantly associated with iron supplementation in anemia hot spot regions of Ethiopia. The odds of receiving iron supplementation among rural resident pregnant women is less likely as compared to pregnant women who live in urban areas. This result aligns with the studies conducted in Tanzania [26], and Dire-Dawa city administration [27]. This is because urban residents have more access to knowledge about the advantages of micronutrients during pregnancy, their availability, and the incentives to use them. Additionally, there are numerous government- and privately-run maternal and child health care services in urban areas where women can easily receive the care they need. In contrast, things are different in rural communities. Moreover, the availability of supplies, including iron supplements, may also play a role in the lower uptake in rural areas. The combination of limited physical access and inadequate supply availability creates obstacles for pregnant women in rural areas to obtain the necessary iron supplements for their health during pregnancy.

Compared to women who had no formal education, pregnant women with greater levels of education are more likely to take iron supplementation. This finding is supported by studies conducted in India [28] and Tanzania [26]. This might be due to educated women generally have greater access to information and possess better research capabilities. Therefore, compared to their less educated counterparts, educated women typically have better access to knowledge and are more capable of looking up the advantages of taking an iron supplement during pregnancy. A pregnant woman whose wealth status is rich and medium level is more likely to take iron supplementation as compared women whose wealth status is poor. The finding of this study is in line with the findings of the studies conducted in East Africa [15, 29, 30]. This might be due to wealthy individuals may have better access to healthcare services, including regular prenatal care, which can increase the likelihood of receiving recommendations and prescriptions for iron supplements to prevent iron deficiency anemia. So, these factors may contribute to higher rates of iron supplement use among wealthier pregnant women compared to those with lower socioeconomic status.

When compared to pregnant women living in households headed by men, those headed by women are less likely to take iron supplements. Because financial hardships are a common issue for pregnant women in households headed by women, which may restrict their access to or ability to purchase iron supplements. And also, due to competing financial demands and lower income levels, pregnant women may choose to put other important need instead of buying iron supplement for their own health. In general, in Ethiopia, women who live in female-headed households often face challenges while seeking healthcare for a number of reasons. These could include lack of finance, transportation, and an inadequate infrastructure for healthcare. Consequently, women who live in female headed households could have challenges while trying to access essential healthcare services including iron supplementation. This is supported by studies [31, 32].

Compared to women who have had more than three children, those who are pregnant but have never given birth are more likely to not take iron supplements. It could be because, in comparison to women who have more than three children, pregnant women without children are more likely to skip out on iron supplements. This is due to a number of factors. First of all, moms who are childless or have fewer children tend to be younger, which may be related to their lack of experience and understanding of the significance of taking iron supplements while pregnant. They might not be as knowledgeable about the advantages and disadvantages of taking iron supplements, which would reduce their propensity to do so [33].

## Strengths and limitations of the study

This study demonstrated strength in its approach by utilizing data from anemia hotspot regions in Ethiopia and employing a multilevel model fitted with the Bayesian approach. This method of analysis allowed for the generation of precise estimates of the parameters, enhancing the accuracy and reliability of the study's findings. By focusing on a specific region with a high prevalence of anemia, the study provided valuable insights into the factors associated with iron supplementation. However, it is important to acknowledge a limitation of the study. Due to data constraints, the researchers were unable to measure the knowledge level of the study participants regarding importance of iron supplementation during pregnancy, other important variables like approaches of health professionals to their client, health education at the community, distance to health facility and media exposure.

## Conclusion and recommendations

The magnitude of iron supplementation in the anemia hotspot regions of Ethiopia is low. Based on the model fitted in Bayesian approach residence, educational level, wealth status, sex

of household head, and number of living children are factors that are significantly associated with iron supplementation in anemia hot spot regions of Ethiopia. Therefore, to increase the magnitude of iron supplementation and to decrease anemia and its complications, the ministry of health and the government of Ethiopia should consider those factors when they design the prevention and control strategies of anemia and its complications. Additionally, future researchers should utilize primary data to assess participants' knowledge about the importance of iron supplementation during pregnancy. They should also consider including other significant variables, such as the approaches of health professionals toward their clients, community health education, distance to health facilities, and media exposure, as these factors significantly influence the level of iron supplementation.

## Acknowledgments

We acknowledge the Demographic and Health Surveys (DHS) center allowing us to access the data set.

## Author Contributions

**Conceptualization:** Yilkal Negesse, Habtamu Temesgen, Wubetu Woyraw, Melsew Setegn Alie, Ayenew Negesse.

**Data curation:** Yilkal Negesse, Habtamu Temesgen, Wubetu Woyraw, Melsew Setegn Alie, Ayenew Negesse.

**Formal analysis:** Yilkal Negesse, Habtamu Temesgen, Wubetu Woyraw, Melsew Setegn Alie.

**Investigation:** Yilkal Negesse, Wubetu Woyraw.

**Methodology:** Yilkal Negesse, Habtamu Temesgen, Wubetu Woyraw.

**Software:** Yilkal Negesse, Habtamu Temesgen.

**Supervision:** Habtamu Temesgen, Ayenew Negesse.

**Validation:** Yilkal Negesse, Melsew Setegn Alie.

**Visualization:** Yilkal Negesse, Ayenew Negesse.

**Writing – original draft:** Yilkal Negesse, Habtamu Temesgen, Wubetu Woyraw, Melsew Setegn Alie, Ayenew Negesse.

**Writing – review & editing:** Yilkal Negesse, Habtamu Temesgen, Wubetu Woyraw, Melsew Setegn Alie, Ayenew Negesse.

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
