## [Decision Letter · Decision Letter 0]

29 Sep 2024

PONE-D-24-30383Magnitude and factors associated with Iron supplementation among pregnant women in Anemia hot spot regions of Ethiopia: Multilevel analysis based on Bayesian approach.PLOS ONE

Dear Dr. Negesse,

Thank you for submitting your manuscript to PLOS ONE. After careful consideration, we feel that it has merit but does not fully meet PLOS ONE’s publication criteria as it currently stands. Therefore, we invite you to submit a revised version of the manuscript that addresses the points raised during the review process.

**ACADEMIC EDITOR: Major revision required**

We look forward to receiving your revised manuscript.

Kind regards,

Alqeer Aliyo Ali, MSc

Academic Editor

PLOS ONE

Additional Editor Comments:

The following issues with the abstract: The abstract requires significant revision, the results in Table 1 should be recalculated, table 3 needs to be rearranged and revised, the discussion also requires extensive revision. I recommend that the author(s) address all the comments made during the review and respond to the reviewers' queries.

Reviewers' comments:

Reviewer's Responses to Questions

**Comments to the Author**

1. Is the manuscript technically sound, and do the data support the conclusions?

Reviewer #1: Yes

Reviewer #2: No

2. Has the statistical analysis been performed appropriately and rigorously? 

Reviewer #1: Yes

Reviewer #2: No

3. Have the authors made all data underlying the findings in their manuscript fully available?

Reviewer #1: Yes

Reviewer #2: Yes

4. Is the manuscript presented in an intelligible fashion and written in standard English?

Reviewer #1: Yes

Reviewer #2: No

5. Review Comments to the Author

Reviewer #1: The authors demonstrate a strong understanding of the research gaps concerning iron supplementation in anemia-prone regions of Ethiopia, and their use of advanced methodology is commendable. However, I have some concerns about the manuscript. Not all findings presented in the results section are discussed in alignment with the study's title or objectives. For example, the discussion lacks coverage of the magnitude of iron supplementation. Additionally, it would be beneficial to highlight the practical implications of these findings for various stakeholders

Reviewer #2: Please remember the following. The abstract' method, results, and conclusion need to be revised. The methodology section, including data analysis, also needs revision. The results and discussion require revision as well. I have attached detailed comments and concerns.

6. PLOS authors have the option to publish the peer review history of their article (what does this mean?). If published, this will include your full peer review and any attached files.

Reviewer #1: No

Reviewer #2: No

---

## [Author Response · Author response to Decision Letter 0]

4 Oct 2024

Date: 02/ 10/ 2024 

To: PLOS ONE

From: Yilkal Negesse

Subject: A letter accompanying first revision in response to Editor’s and reviewer’s comments and questions 

Dear Editor and reviewers 

I would like to express my gratitude to you and the reviewers for your valuable feedback on our manuscript titled “Magnitude and factors associated with Iron supplementation among pregnant women in Anemia hot spot regions of Ethiopia: Multilevel analysis based on Bayesian approach." We appreciate the time and effort taken to review our work and provide insightful comments.

In response to the reviewers' questions and suggestions, we have made several revisions to enhance the clarity and quality of the manuscript. Below, I outline our responses to the specific comments and questions raised:

With regards 

Yilkal Negesse

On the behalf of other authors

Editor and reviewers’ comments and questions 

Author’s response: Thank you very much for your comment. Noted and revised accordingly

2. We note that you have indicated that there are restrictions to data sharing for this study.

Author’s response: Thank you for your comment. The data utilized in this study is secondary data that is openly accessible. Therefore, we can share it upon request.

Author’s response: Thank you for your comment. Noted and corrected accordingly.

Abstract 

1. The author should mention the level of statistical significance in the methods section of the abstract.

Author’s response: Thank you very much for your comment. We have noted and corrected it as indicated between lines 40 and 44

2. The data analysis model should also be mentioned.

Author’s response: Thank you very much for your comment. As indicated from line 39, we have noted and corrected it to read: “A multilevel binary logistic regression model based on the Bayesian approach was fitted using the Brms R package to identify the determinants of iron supplementation in anemia hotspot regions of Ethiopia.”

3. The result of the abstract states that the overall magnitude of iron supplementation among pregnant women is 55.5%, but it requires a confidence interval.

Author’s response: thank you very much for your comment. Noted and corrected it accordingly. 

4. The conclusion of the abstract overlaps with the result. The author should rewrite it to avoid repetition.

Author’s response: thank you very much for your comment. Noted and revised it accordingly. 

5. When stating that iron supplementation among pregnant women in anemia hotspot regions of Ethiopia is low, the author should clarify how "low" is determined. It would be helpful to compare this magnitude with national or global levels for context.

Author’s response: Thank you very much for your comment. According to the World Health Organization (WHO), daily iron and folic acid supplementation is recommended for all pregnant women for at least 90 days to prevent maternal anemia and related complications. While the WHO guidelines and other documents do not define specific thresholds for "low" or "high" coverage of iron supplementation, they emphasize that all pregnant women should receive this intervention. Given this context, nearly half of the pregnant women in anemia hotspot regions of Ethiopia do not receive iron supplementation, leading us to conclude that the coverage is indeed low.

Methodology

6. It is unclear, 2116 was obtained out of how much overall data?

Author’s response: thank you very much for your question. In this study, we utilized the 2019 Ethiopia Demographic and Health Survey (EDHS) dataset, specifically the Individual Recode (IR) file, which includes information on women of reproductive age (15-49 years). At national level, a total of 8,885 women were selected, of whom 3,908 were eligible to respond to questions regarding iron supplementation and 4,977 were not. However, our study specifically targeted anemia hotspot regions in Ethiopia, resulting in a final sample size of 2,116 women.

7. The method of data extraction is not clear.

Author’s response: Thank you very much for your comment. We have noted and corrected it accordingly. 

8. The process of data entry and analysis is not clearly explained. It would be helpful to specify the software used for data entry and analysis.

Thank you for your comment. As outlined in the Data Source subsection of the manuscript, the data used in this study is secondary and was obtained in STATA format after securing permission via an online request. Since the data was pre-collected, no additional data entry was required. But for the data analysis, we used STATA software for descriptive statistical analysis and R software for inferential statistical analysis (to fit multilevel binary logistic regression model based on Bayesian approach). This is noted and corrected accordingly. 

9. The methodology should explain how confounders were controlled.

Author’s response: Thank you very much for your comment. We have noted and corrected it accordingly. 

Result 

10. The author needs to describe the findings using the figure (number) with the percentage. For example, an overall magnitude of 55.5% (?/2116). Describing the finding only in percent in the main result section is not acceptable

Thank you so much for your comment. Noted and corrected it accordingly.

11. A confidence interval for the overall magnitude of 55.5% is required.

Thank you so much for your comment. Noted and corrected it accordingly

12. In the table, the overall frequency of residence is (867+308=1175), age group (521+520+135=1176), marital status (1087+14+24+10+42=1,177), and sex of household head (886+289=175). It's important to investigate the reasons for the overall iron supplement variations among variables. This indicates a false report, so the author should recalculate it.

Thank you very much for your insightful comment and question. I acknowledge that I was a bit confused to report it. After weighting and reviewing the data, I noticed that many figures were in decimals. According to many scholars, it is generally preferred to round population figures up to the next whole number, as decimals are not typically used when describing populations. Based on this approach, I rounded up the figures. So, the reason of the variation is this one. But, if you suggest me an idea different from this, I am eager to accept and to revise the document. For your reference, the STATA output looks like this:

13. Table 3 is confusing, and unnecessary analysis was presented. It's unclear why the author included the results of Estimates, SE, Tail_ESS, and Bulk_ESS in the table, as they were not used in the discussion and conclusion. All those columns should be removed. I recommend that the author revise the entire Table 3 and present it as follows:

Variables 

Category 

Iron supplement 

COR (95% Cl) 

AOR (95% Cl) 

P- value

 Yes N (%) No N (%) 

Residence Urban ?(?) ?(?) 1 1 

 Rural ?(?) ?(?) 0.16[0.04-0.63] 0.16[0.04-0.63] 0.002

Thank you for your valuable insights and comments. However, since this study is conducted based on Bayesian approach, using Crude Odds Ratios (COR) and p-values is not appropriate, as these are typically used in frequentist approaches. Bayesian analysis focuses on adjusted odds ratios (AOR), credible intervals (CrI), and the convergence of the algorithm. As explained in the manuscript, results from the posterior distribution are only reliable once the chain has reached convergence, or its stationary distribution. To ensure that the results are from a properly converged posterior distribution, it is necessary to report metrics such as SE, Tail_ESS, Bulk_ESS, and Rhat in the regression table. These measures ensure the robustness of the Bayesian estimates. Therefore, we report it in that way.

Discussion

14. There is no discussion about the magnitude of the iron supplement, which is the main objective. I strongly recommend that the author should discuss the findings of the magnitude.

Thank you so much for your comment. Noted and corrected it accordingly.

15. In the discussion section, the author only described the study result but did not discuss or compare it with previous study findings. It should be revised.

Thank you so much for your comment. Noted and revised it accordingly.

16. In paragraph 2 of the discussion, it is stated "This finding is supported by studies (25-27)." This is an incomplete sentence. It should be mentioned where these studies were conducted and what they reported.

 Thank you so much for your comment. Noted and revised it accordingly.

17. In paragraph 3, it is stated “This finding is supported by studies conducted in India and Tanzania (25, 28).” However, reference 25 does not indicate a study conducted in India, and 28 does not indicate a study in Tanzania. I strongly recommend that the author should cite the references in the correct place; otherwise, this paper is unacceptable.

Thank you for your insightful feedback. We noticed that we had mistakenly changed the reference numbers, assigning 25 to India and 28 to Tanzania, when it should have been the reverse. We have now corrected the references accordingly.

Conclusion and Recommendation

18. Since there are many limitations of the study, the author should recommend future research to address these gaps.

Thank you so much for your comment. Noted and corrected accordingly.

---

## [Decision Letter · Decision Letter 1]

18 Oct 2024

Magnitude and factors associated with Iron supplementation among pregnant women in Anemia hot spot regions of Ethiopia: Multilevel analysis based on Bayesian approach.

PONE-D-24-30383R1

Dear Dr. Negesse,

We’re pleased to inform you that your manuscript has been judged scientifically suitable for publication and will be formally accepted for publication once it meets all outstanding technical requirements.

Kind regards,

Alqeer Aliyo Ali, MSc

Academic Editor

PLOS ONE

Additional Editor Comments (optional):

Reviewers' comments:

Reviewer's Responses to Questions

**Comments to the Author**

1. If the authors have adequately addressed your comments raised in a previous round of review and you feel that this manuscript is now acceptable for publication, you may indicate that here to bypass the “Comments to the Author” section, enter your conflict of interest statement in the “Confidential to Editor” section, and submit your "Accept" recommendation.

Reviewer #2: All comments have been addressed

2. Is the manuscript technically sound, and do the data support the conclusions?

Reviewer #2: Yes

3. Has the statistical analysis been performed appropriately and rigorously? 

Reviewer #2: Yes

4. Have the authors made all data underlying the findings in their manuscript fully available?

Reviewer #2: Yes

5. Is the manuscript presented in an intelligible fashion and written in standard English?

Reviewer #2: Yes

6. Review Comments to the Author

Reviewer #2: The authors have improved the manuscript accordingly. I hope the manuscript is suitable for publication in the journal.

7. PLOS authors have the option to publish the peer review history of their article (what does this mean?). If published, this will include your full peer review and any attached files.

Reviewer #2: No

---

## [Editor Report · Acceptance letter]

24 Oct 2024

PONE-D-24-30383R1 

PLOS ONE

Dear Dr. Negesse, 

I'm pleased to inform you that your manuscript has been deemed suitable for publication in PLOS ONE. Congratulations! Your manuscript is now being handed over to our production team.

Kind regards, 

on behalf of

Mr. Alqeer Aliyo Ali 

Academic Editor

PLOS ONE